# DropoutNet: Addressing Cold Start in Recommender Systems

**Maksims Volkovs**
layer6.ai
maks@layer6.ai

**Guangwei Yu**
layer6.ai
guang@layer6.ai

**Tomi Poutanen**
layer6.ai
tomi@layer6.ai

## Abstract

Latent models have become the default choice for recommender systems due to their performance and scalability. However, research in this area has primarily focused on modeling user-item interactions, and few latent models have been developed for cold start. Deep learning has recently achieved remarkable success showing excellent results for diverse input types. Inspired by these results we propose a neural network based latent model called DropoutNet to address the cold start problem in recommender systems. Unlike existing approaches that incorporate additional content-based objective terms, we instead focus on the optimization and show that neural network models can be explicitly trained for cold start through dropout. Our model can be applied on top of any existing latent model effectively providing cold start capabilities, and full power of deep architectures. Empirically we demonstrate state-of-the-art accuracy on publicly available benchmarks. Code is available at `https://github.com/layer6ai-labs/DropoutNet`.

## 1   Introduction

Popularity of online content delivery services, e-commerce, and social web has highlighted an important challenge of surfacing relevant content to consumers. Recommender systems have proven to be effective tools for this task, receiving increasingly more attention. One common approach to building accurate recommender models is collaborative filtering (CF). CF is a method of making predictions about an individual's preferences based on the preference information from other users. CF has been shown to work well across various domains [19], and many successful web-services such as Netflix, Amazon and YouTube use CF to deliver highly personalized recommendations to their users.

The majority of the existing approaches in CF can be divided into two categories: neighbor-based and model-based. Model-based approaches, and in particular latent models, are typically the preferred choice since they build compact representations of the data and achieve high accuracy. These representations are optimized for fast retrieval and can be scaled to handle millions of users in real-time. For these reasons we concentrate on latent approaches in this work. Latent models are typically learned by applying a variant of low rank approximation to the target preference matrix. As such, they work well when lots of preference information is available but start to degrade in highly sparse settings. The most extreme case of sparsity known as *cold start* occurs when no preference information is available for a given user or item. In such cases, the only way a personalized recommendation can be generated is by incorporating additional content information. Base latent approaches cannot incorporate content, so a number of hybrid models have been proposed [3, 21, 22] to combine preference and content information. However, most hybrid methods introduce additional objective terms considerably complicating learning and inference. Moreover, the content part of the objective is typically generative [21, 9, 22] forcing the model to "explain" the content rather than use it to maximize recommendation accuracy.

Recently, deep learning has achieved remarkable success in areas such as computer vision [15, 11], speech [12, 10] and natural language processing [5, 16]. In all of these areas end-to-end deep neu-

ral network (DNN) models achieve state-of-the-art accuracy with virtually no feature engineering. These results suggest that deep learning should also be highly effective at modeling content for recommender systems. However, while there has been some recent progress in applying deep learning to CF [7, 22, 6, 23], little investigation has been done on using deep learning to address the cold start problem.

In this work we propose a model to address this gap. Our approach is based on the observation that cold start is equivalent to the missing data problem where preference information is missing. Hence, instead of adding additional objective terms to model content, we modify the learning procedure to explicitly condition the model for the missing input. The key idea behind our approach is that by applying dropout [18] to input mini-batches, we can train DNNs to generalize to missing input. By selecting an appropriate amount of dropout we show that it is possible to learn a DNN-based latent model that performs comparably to state-of-the-art on warm start while significantly outperforming it on cold start. The resulting model is simpler than most hybrid approaches and uses a single objective function, jointly optimizing all components to maximize recommendation accuracy.

An additional advantage of our approach is that it can be applied on top of any existing latent model to provide/enhance its cold start capability. This requires virtually no modification to the original model thus minimizing the implementation barrier for any production environment that's already running latent models. In the following sections we give a detailed description of our approach and show empirical results on publicly available benchmarks.

## 2   Framework

In a typical CF problem we have a set of $N$ users $\mathcal{U} = \{u_1, ..., u_N\}$ and a set of $M$ items $\mathcal{V} = \{v_1, ..., v_M\}$. The users' feedback for the items can be represented by an $N \times M$ preference matrix $\mathbf{R}$ where $\mathbf{R}_{uv}$ is the preference for item $v$ by user $u$. $\mathbf{R}_{uv}$ can be either explicitly provided by the user in the form of rating, like/dislike etc., or inferred from implicit interactions such as views, plays and purchases. In the explicit setting $\mathbf{R}$ typically contains graded relevance (e.g., 1-5 ratings), while in the implicit setting $\mathbf{R}$ is often binary; we consider both cases in this work. When no preference information is available $\mathbf{R}_{uv} = 0$. We use $\mathcal{U}(v) = \{u \in \mathcal{U} \mid \mathbf{R}_{uv} \neq 0\}$ to denote the set of users that expressed preference for $v$, and $\mathcal{V}(u) = \{v \in \mathcal{V} \mid \mathbf{R}_{uv} \neq 0\}$ to denote the set of items that $u$ expressed preference for. In cold start no preference information is available and we formally define cold start when $\mathcal{V}(u) = \emptyset$ and $\mathcal{U}(v) = \emptyset$ for a given user $u$ and item $v$.

Additionally, in many domains we often have access to content information for both users and items. For items, this information can come in the form of text, audio or images/video. For users we could have profile information (age, gender, location, device etc.), and social media data (Facebook, Twitter etc.). This data can provide highly useful signal for recommender models, and is particularly effective in sparse and cold start settings where little or no preference information is available. After applying relevant transformations most content information can be represented by fixed-length feature vectors. We use $\mathbf{\Phi}^{\mathcal{U}}$ and $\mathbf{\Phi}^{\mathcal{V}}$ to denote the content features for users and items respectively where $\mathbf{\Phi}_u^{\mathcal{U}}$ ($\mathbf{\Phi}_v^{\mathcal{V}}$) is the content feature vector for user $u$ (item $v$). When content is missing the corresponding feature vector is set to 0. The goal is to use the preference information $\mathbf{R}$ together with content $\mathbf{\Phi}^{\mathcal{U}}$ and $\mathbf{\Phi}^{\mathcal{V}}$, to learn accurate and robust recommendation model. Ideally this model should handle all stages of the user/item journey: from cold start, to early stage sparse preferences, to a late stage well-defined preference profile.

## 3   Relevant Work

A number of hybrid latent approaches have been proposed to address cold start in CF. One of the more popular models is the collaborative topic regression (CTR) [21] which combines latent Dirichlet allocation (LDA) [4] and weighted matrix factorization (WMF) [13]. CTR interpolates between LDA representations in cold start and WMF when preferences are available. Recently, several related approaches have been proposed. Collaborative topic Poisson factorization (CTPF) [8] uses a similar interpolation architecture but replaces both LDA and WMF components with Poisson factorization [9]. Collaborative deep learning (CDL) [22] is another approach with analogous architecture where LDA is replaced with a stacked denoising autoencoder [20].

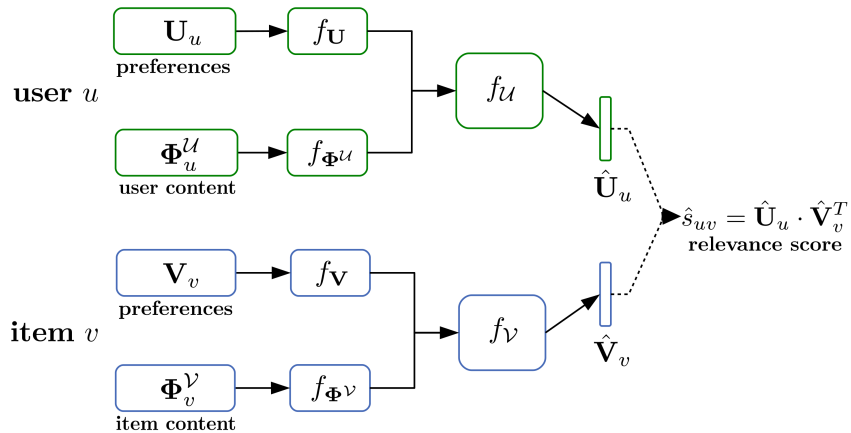

Figure 1: DropoutNet architecture diagram. For each user $u$, the preference $\mathbf{U}_u$ and content $\mathbf{\Phi}_u^{\mathcal{U}}$ inputs are first passed through the corresponding DNNs $f_{\mathbf{U}}$ and $f_{\mathbf{\Phi}^{\mathcal{U}}}$. Top layer activations are then concatenated together and passed to the fine-tuning network $f_{\mathcal{U}}$ which outputs the latent representation $\hat{\mathbf{U}}_u$. Items are handled in a similar fashion with $f_{\mathbf{V}}$, $f_{\mathbf{\Phi}^{\mathcal{V}}}$ and $f_{\mathcal{V}}$ to produce $\hat{\mathbf{V}}_v$. All components are optimized jointly with back-propagation and then kept fixed during inference. Retrieval is done in the new latent space using $\hat{\mathbf{U}}$ and $\hat{\mathbf{V}}$ that replace the original representations $\mathbf{U}$ and $\mathbf{V}$.

While these models achieve highly competitive performance, they also share several disadvantages. First, they incorporate both preference and content components into the objective function making it highly complex. CDL for example, contains four objective terms and requires tuning three combining weights in addition to WMF and autoencoder parameters. This makes it challenging to tune these models on large datasets where every parameter setting experiment is expensive and time consuming. Second, the formulation of each model assumes cold start items and is not applicable to cold start users. Most online services have to frequently incorporate new users and items and thus require models that can handle both. In principle it is possible to derive an analogous model for users and jointly optimize both models. However, this would require an even more complex objective nearly doubling the number of free parameters. One of the main questions that we aim to address with this work is whether we develop a simpler cold start model that is applicable to both users and items?

In addition to CDL, a number of approaches haven been proposed to leverage DNNs for CF. One of the earlier approaches DeepMusic [7] aimed to predict latent representations learned by a latent model using content only DNN. Recently, [6] described YouTube's two-stage recommendation model that takes as input user session (recent plays and searches) and profile information. Latent representations for items in a given session are averaged, concatenated with profile information, and passed to a DNN which outputs a session-dependent latent representation for the user. Averaging the items addresses variable length input problem but can loose temporal aspects of the session. To more accurately model how users' preferences change over time a recurrent neural network (RNN) approach has been proposed by [23]. RNN is applied sequentially to one item at a time, and after all items are processed hidden layer activations are used as latent representation.

Many of these models show clear benefits of applying deep architectures to CF. However, few investigate cold start and sparse setting performance when content information is available. Arguably, we expect deep learning to be the most beneficial in these scenarios due to its excellent generalization to various content types. Our proposed approach aims to leverage this advantage and is most similar to [6]. We also use latent representations as preference feature input for users and items, and combine them with content to train a hybrid DNN-based model. But unlike [6] which focuses primarily on warm start users, we develop analogous models for both users and items, and then show how these models can be trained to explicitly handle cold start.

## 4   Our Approach

In this section we describe the architecture of our model that we call DropoutNet, together with learning and inference procedures. We begin with input representation. Our aim is to develop a model that is able to handle both cold and warm start scenarios. Consequently, input to the model

needs to contain content and preference information. One option is to directly use rows and columns of $\mathbf{R}$ in their raw form. However, these become prohibitively large as the number of users and items grows. Instead, we take a similar approach to [6] and [23], and use latent representations as preference input. Latent models typically approximate the preference matrix with a product of low rank matrices $\mathbf{U}$ and $\mathbf{V}$:

$$\mathbf{R}_{uv} \approx \mathbf{U}_u \mathbf{V}_v^T \qquad (1)$$

where $\mathbf{U}_u$ and $\mathbf{V}_v$ are the latent representations for user $u$ and item $v$ respectively. Both $\mathbf{U}$ and $\mathbf{V}$ are dense and low dimensional with rank $D \ll \min(N, M)$. Noting the strong performance of latent approaches on a wide range of CF datasets, it is adequate to assume that the latent representations accurately summarize preference information about users and items. Moreover, low input dimensionality significantly reduces model complexity for DNNs since activation size of the first hidden layer is directly proportional to the input size. Given these advantages we set the input to $[\mathbf{U}_u, \mathbf{\Phi}_u^{\mathcal{U}}]$ and $[\mathbf{V}_u, \mathbf{\Phi}_v^{\mathcal{V}}]$ for each user $u$ and item $v$ respectively.

## 4.1 Model Architecture

Given the joint preference-content input we propose to apply a DNN model to map it into a new latent space that incorporates both content and preference information. Formally, preference $\mathbf{U}_u$ and content $\mathbf{\Phi}_u^{\mathcal{U}}$ inputs are first passed through the corresponding DNNs $f_{\mathbf{U}}$ and $f_{\mathbf{\Phi}^{\mathcal{U}}}$. Top layer activations are then concatenated together and passed to the fine-tuning network $f_{\mathcal{U}}$ which outputs the latent representation $\hat{\mathbf{U}}_u$. Items are handled in a similar fashion with $f_{\mathbf{V}}$, $f_{\mathbf{\Phi}^{\mathcal{V}}}$ and $f_{\mathcal{U}}$ to produce $\hat{\mathbf{V}}_v$. We use separate components for preference and content inputs to handle complex structured content such as images that can't be directly concatenated with preference input in raw form. Another advantage of using a split architecture is that it allows to use any of the publicly available (or proprietary) pre-trained models for $f_{\mathbf{\Phi}^{\mathcal{U}}}$ and/or $f_{\mathbf{\Phi}^{\mathcal{V}}}$. Training can then be significantly accelerated by updating only the last few layers of each pre-trained network. For domains such as vision where models can exceed 100 layers [11], this can effectively reduce the training time from days to hours. Note that when content input is "compatible" with preference representations we remove $f_{\mathbf{U}}$ and $f_{\mathbf{\Phi}^{\mathcal{U}}}$, and directly apply $f_{\mathcal{U}}$ to concatenated input $[\mathbf{U}_u, \mathbf{\Phi}_u^{\mathcal{U}}]$. To avoid notation clutter we omit the sub-networks and use $f_{\mathcal{U}}$ and $f_{\mathcal{V}}$ to denote user and item models in subsequent sections.

During training all components are optimized jointly with back-propagation. Once the model is trained we fix it, and make forward passes to map $\mathbf{U} \to \hat{\mathbf{U}}$ and $\mathbf{V} \to \hat{\mathbf{V}}$. All retrieval is then done using $\hat{\mathbf{U}}$ and $\hat{\mathbf{V}}$ with relevance scores estimated as before by $\hat{s}_{uv} = \hat{\mathbf{U}}_u \hat{\mathbf{V}}_v^T$. Figure 1 shows the full model architecture with both user and item components.

## 4.2 Training For Cold Start

During training we aim to generalize the model to cold start while preserving warm start accuracy. We discussed that existing hybrid model approach this problem by adding additional objective terms and training the model to fall-back on content representations when preferences are not available. However, this complicates learning by forcing the implementer to balance multiple objective terms in addition to training content representations. Moreover, content part of the objective is typically generative forcing the model to explain the observed data instead of using it to maximize recommendation accuracy. This can waste capacity by modeling content aspects that are not useful for recommendations.

We take a different approach and borrow ideas from denoising autoencoders [20] by training the model to reconstruct the input from its corrupted version. The goal is to learn a model that would still produce accurate representations when parts of the input are missing. To achieve this we propose an objective to reproduce the relevance scores after the input is passed through the model:

$$\mathcal{O} = \sum_{u,v} (\mathbf{U}_u \mathbf{V}_v^T - f_{\mathcal{U}}(\mathbf{U}_u, \mathbf{\Phi}_u^{\mathcal{U}}) f_{\mathcal{V}}(\mathbf{V}_v, \mathbf{\Phi}_v^{\mathcal{V}})^T)^2 = \sum_{u,v} (\mathbf{U}_u \mathbf{V}_v^T - \hat{\mathbf{U}}_u \hat{\mathbf{V}}_v^T)^2 \qquad (2)$$

$\mathcal{O}$ minimizes the difference between scores produced by the input latent model and DNN. When all input is available this objective is trivially minimized by setting the content weights to $0$ and learning identity function for preference input. This is a desirable property for reasons discussed below.

In cold start either $\mathbf{U}_u$ or $\mathbf{V}_v$ (or both) is missing so our main idea is to train for this by applying *input dropout* [18]. We use stochastic mini-batch optimization and randomly sample user-item pairs

to compute gradients and update the model. In each mini-batch a fraction of users and items is selected at random and their preference inputs are set to 0 before passing the mini-batch to the model. For "dropped out" pairs the model thus has to reconstruct the relevance scores without seeing the preference input:

$$\text{user cold start:} \quad \mathcal{O}_{uv} = (\mathbf{U}_u \mathbf{V}_v^T - f_{\mathcal{U}}(\mathbf{0}, \mathbf{\Phi}_u^{\mathcal{U}}) f_{\mathcal{V}}(\mathbf{V}_v, \mathbf{\Phi}_v^{\mathcal{V}})^T)^2$$
$$\text{item cold start:} \quad \mathcal{O}_{uv} = (\mathbf{U}_u \mathbf{V}_v^T - f_{\mathcal{U}}(\mathbf{U}_u, \mathbf{\Phi}_u^{\mathcal{U}}) f_{\mathcal{V}}(\mathbf{0}, \mathbf{\Phi}_v^{\mathcal{V}})^T)^2$$

(3)

Training with dropout has a two-fold effect: pairs with dropout encourage the model to only use content information, while pairs without dropout encourage it to ignore content and simply reproduce preference input. The net effect is balanced between these two extremes. The model learns to reproduce the accuracy of the input latent model when preference data is available while also generalizing to cold start. Dropout thus has a similar effect to hybrid preference-content interpolation objectives but with a much simpler architecture that is easy to optimize. An additional advantage of using dropout is that it was originally developed as a way of regularizing the model. We observe a similar effect here, finding that additional regularization is rarely required even for deeper and more complex models.

There are interesting parallels between our model and areas such as denoising autoencoders [20] and dimensionality reduction [17]. Analogous to denoising autoencoders, our model is trained to reproduce the input from a noisy version. The noise comes in the form of dropout that fully removes a subset of input dimensions. However, instead of reconstructing the actual uncorrupted input we minimize pairwise distances between points in the original and reconstructed spaces. Considering relevance scores $S = \{\mathbf{U}_u \mathbf{V}_v^T \mid u \in \mathcal{U}, v \in \mathcal{V}\}$ and $\hat{S} = \{\hat{\mathbf{U}}_u \hat{\mathbf{V}}_v^T \mid u \in \mathcal{U}, v \in \mathcal{V}\}$ as sets of points in one dimensional space, the goal is to preserve the relative ordering between the points in $\hat{S}$ produced by our model and the original set $S$. We focus on reconstructing distances because it gives greater flexibility allowing the model to learn an entirely new latent space, and not tying it to a representation learned by another model. This objec-

---

**Algorithm 1: Learning Algorithm**

**Input:** $\mathbf{R}$, $\mathbf{U}$, $\mathbf{V}$, $\mathbf{\Phi}^{\mathcal{U}}$, $\mathbf{\Phi}^{\mathcal{V}}$
**Initialize:** user model $f_{\mathcal{U}}$, item model $f_{\mathcal{V}}$
**repeat** {DNN optimization}
    sample mini-batch $B = \{(u_1, v_1), ..., (u_k, v_k)\}$
    **for** each $(u, v) \in B$ **do**
        apply one of:
            1. **leave as is**
            2. **user dropout:**
                $[\mathbf{U}_u, \mathbf{\Phi}_u^{\mathcal{U}}] \to [\mathbf{0}, \mathbf{\Phi}_u^{\mathcal{U}}]$
            3. **item dropout:**
                $[\mathbf{V}_v, \mathbf{\Phi}_v^{\mathcal{V}}] \to [\mathbf{0}, \mathbf{\Phi}_v^{\mathcal{V}}]$
            4. **user transform:**
                $[\mathbf{U}_u, \mathbf{\Phi}_u^{\mathcal{U}}] \to [\text{mean}_{v \in \mathcal{V}(u)} \mathbf{V}_v, \mathbf{\Phi}_u^{\mathcal{U}}]$
            5. **item transform:**
                $[\mathbf{V}_v, \mathbf{\Phi}_v^{\mathcal{V}}] \to [\text{mean}_{u \in \mathcal{V}(v)} \mathbf{U}_u, \mathbf{\Phi}_v^{\mathcal{V}}]$
    **end for**
    update $f_{\mathcal{V}}$, $f_{\mathcal{U}}$ using $B$
**until** convergence
**Output:** $f_{\mathcal{V}}$, $f_{\mathcal{U}}$

---

tive is analogous to many popular dimensionality reduction models that project the data to a low dimensional space where relative distances between points are preserved [17]. In fact, many of the objective functions developed for dimensionality reduction can also be used here.

A drawback of the objective in Equation 2 is that it depends on the input latent model and thus its accuracy. However, empirically we found this objective to work well producing robust models. The main advantages are that, first, it is simple to implement and has no additional free parameters to tune making it easy to apply to large datasets. Second, in mini-batch mode, $NM$ unique user-item pairs can be sampled to update the networks. Even for moderate size datasets the number of pairs is in the billions making it significantly easier to train large DNNs without over-fitting. The performance is particularly robust on sparse implicit datasets commonly found in CF where $\mathbf{R}$ is binary and over 99% sparse. In this setting training with mini-batches sampled from raw $\mathbf{R}$ requires careful tuning to avoid oversampling 0's, and to avoid getting stuck in bad local optima.

## 4.3 Inference

Once training is completed, we fix the model and make forward passes to infer new latent representations. Ideally we would apply the model continuously throughout all stages of the user (item) journey – starting from cold start, to first few interactions and finally to an established preference profile. However, to update latent representation $\hat{\mathbf{U}}_u$ as we observe first preferences from a cold

start user $u$, we need to infer the input preference vector $\mathbf{U}_u$. As many leading latent models use complex non-convex objectives, updating latent representations with new preferences is a non-trivial task that requires iterative optimization. To avoid this we use a simple trick by representing each user as a weighted sum of items that the user interacted with until the input latent model is retrained. Formally, given cold start user $u$ that has generated new set of interactions $\mathcal{V}(u)$ we approximate $\mathbf{U}_u$ with the average latent representations of the items in $\mathcal{V}(u)$:

$$\mathbf{U}_u \approx \frac{1}{|\mathcal{V}(u)|} \sum_{v \in \mathcal{V}(u)} \mathbf{V}_v \qquad (4)$$

Using this approximation, we then make a forward pass through the user DNN to get the updated representation: $\hat{\mathbf{U}}_u = f_{\mathcal{U}}(\text{mean}_{v \in \mathcal{V}(u)} \mathbf{V}_v, \boldsymbol{\Phi}_u^{\mathcal{U}})$. This procedure can be used continuously in near real-time as new data is collected until the input latent model is re-trained. Cold start items are handled in a similar way by using averages of user representations. Distribution of representations obtained via this approximation can deviate from the one produced by the input latent model. We explicitly train for this using a similar idea to dropout for cold start. Throughout learning preference input for a randomly chosen subset of users and items in each mini-batch is replaced with Equation 4. We alternate between dropout and this transformation and control for the relative frequency of each transformation (i.e., dropout fraction). Algorithm 1 outlines the full learning procedure.

## 5   Experiments

To validate the proposed approach, we conducted extensive experiments on two publicly available datasets: CiteULike [21] and the ACM RecSys 2017 challenge dataset [2]. These datasets are chosen because they contain content information, allowing cold start evaluation. We implemented Algorithm 1 using the TensorFlow library [1]. All experiments were conducted on a server with 20-core Intel Xeon CPU E5-2630 CPU, Nvidia Titan X GPU and 128GB of RAM. We compare our model against leading CF approaches including WMF [13], CTR [21], DeepMusic [7] and CDL [22] described in Section 3. For all baselines except DeepMusic, we use the code released by respective authors, and extensively tune each model to find an optimal setting of hyper-parameters. For DeepMusic we use a modified version of the model replacing the objective function from [7] with Equation 2 which we found to work better. To make comparison fair we use the same DNN architecture (number of hidden layers and layer size) for DeepMusic and our models.

All DNN models are trained with mini batches of size 100, fixed learning rate and momentum of 0.9. Algorithm 1 is applied directly to the mini batches, and we alternate between applying dropout, and inference transforms. Using $\tau$ to denote the dropout rate, for each batch we randomly select $\tau * batch\_size$ users and items. Then for batch 1 we apply dropout to selected users and items, for batch 2 inference transform and so on. We found this procedure to work well across different datasets and use it in all experiments.

### 5.1   CiteULike

At CiteULike, registered users create scientific article libraries and save them for future reference. The goal is to leverage these libraries to recommend relevant new articles to each user. We use a subset of the CiteULike data with 5,551 users, 16,980 articles and 204,986 observed user-article pairs. This is a binary problem with $\mathbf{R}(u, v) = 1$ if article $v$ is in $u$'s library and $\mathbf{R}(u, v) = 0$ otherwise. $\mathbf{R}$ is over 99.8% sparse with each user collecting an average of 37 articles. In addition to preference data, we also have article content information in the form of title and abstract. To make the comparison fair we follow the approach of [21] and use the same vocabulary of top 8,000 words selected by tf-idf. This produces the $16,980 \times 8,000$ item content matrix $\boldsymbol{\Phi}^{\mathcal{V}}$; since no user content is available $\boldsymbol{\Phi}^{\mathcal{U}}$ is dropped from the model. For all evaluation we use Fold 1 from [21] (results on other folds are nearly identical) and report results of the test set from this fold. We modify warm start evaluation and measure accuracy by generating recommendations from the full set of $16,980$ articles for each user (excluding training interactions). This makes the problem more challenging, and provides a better evaluation of model performance. Cold start evaluation is the same as in [21], we remove a subset of 3396 articles from the training data and then generate recommendations from these articles at test time.

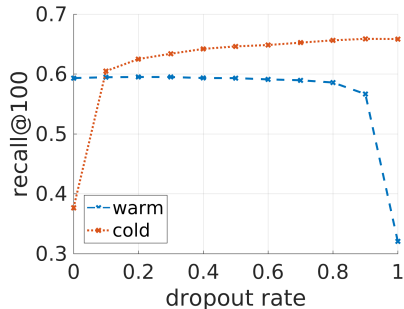

Figure 2: CiteULike warm and cold start results for dropout rates between 0 and 1.

| Method | Warm Start | Cold Start |
|---|---|---|
| WMF [13] | 0.592 | . |
| CTR [21] | 0.597 | 0.589 |
| DeepMusic [7] | 0.371 | 0.601 |
| CDL [22] | **0.603** | 0.573 |
| DN-WMF | 0.593 | **0.636** |
| DN-CDL | 0.598 | 0.629 |

Table 1: CiteULike recall@100 warm and cold start test set results.

We fix rank $D = 200$ for all models to stay consistent with the setup used in [21]. For our model we found that 1-hidden layer architectures with 500 hidden units and tanh activations gave good performance and going deeper did not significantly improve results. To train the model for cold start we apply dropout to preference input as outlined in Section 4.2. Here, we only apply dropout to item preferences since only item content is available. Figure 2 shows warm and cold start recall@100 accuracy for dropout rate (probability to drop) between 0 and 1. From the figure we see an interesting pattern where warm start accuracy remains virtually unchanged decreasing by less than 1% until dropout reaches 0.7 where it rapidly degrades. Cold start accuracy on the other hand, steadily increases with dropout. Moreover, without dropout cold start performance is poor and even dropout of 0.1 improves it by over 60%. This indicates that there is a region of dropout values where significant gains in cold start accuracy can be achieved *without* losses on warm start. Similar patterns were observed on other datasets and further validate that the proposed approach of applying dropout for cold start generalization achieves the desired effect.

Warm and cold start recall@100 results are shown in Table 1. To verify that our model can be trained in conjunction with any existing latent model, we trained two versions denoted DN-WMF and DN-CDL, that use WMF and CDL as input preference models respectively. Both models were trained with preference input dropout rate of 0.5. From the table we see that most baselines produce similar results on warm start which is expected since virtually all of these models use WMF objective to model $\mathbf{R}$. One exception is DeepMusic that performs significantly worse than other baselines. This can be attributed to the fact that in DeepMusic item latent representations are functions of content only and thus lack preference information. DN-WMF and DN-CDL on the other hand, perform comparably to the best baseline indicating that adding preference information as input into the model significantly improves performance over content only models like DeepMusic. Moreover, as Figure 2 suggests even aggressive dropout of 0.5 does not affect warm start performance and the our model is still able to recover the accuracy of the input latent model.

Cold start results are more diverse, as expected best cold start baseline is DeepMusic. Unlike CTR and CDL that have unsupervised and semi-supervised content components, DeepMusic is end-to-end supervised, and can thus learn representations that are better tailored to the target retrieval task. We also see that DNN-WMF outperforms all baselines improving recall@100 by 6% over the best baseline. This indicates that incorporating preference information as input during training can also improve cold start generalization. Moreover, WMF can't be applied to cold start so our model effectively adds cold start capability to WMF with excellent generalization and without affecting performance on warm start. Similar pattern can be seen for DN-CDL that improves cold start performance of CDL by almost 10% without affecting warm start.

## 5.2 RecSys

The ACM RecSys 2017 dataset was released as part of the ACM RecSys 2017 Challenge [2]. It's a large scale data collection of user-job interactions from the career oriented social network XING (European analog of LinkedIn). Importantly, this is one of the only publicly available datasets that contains both user and item content information enabling cold start evaluation on both. In total there are 1.5M users, 1.3M jobs and over 300M interactions. Interactions are divided into six types {*impression, click, bookmark, reply, delete, recruiter*}, and each interaction is recorded with the corresponding type and timestamp. In addition, for users we have access to profile information such as education, work experience, location and current position. Similarly, for items we have industry,

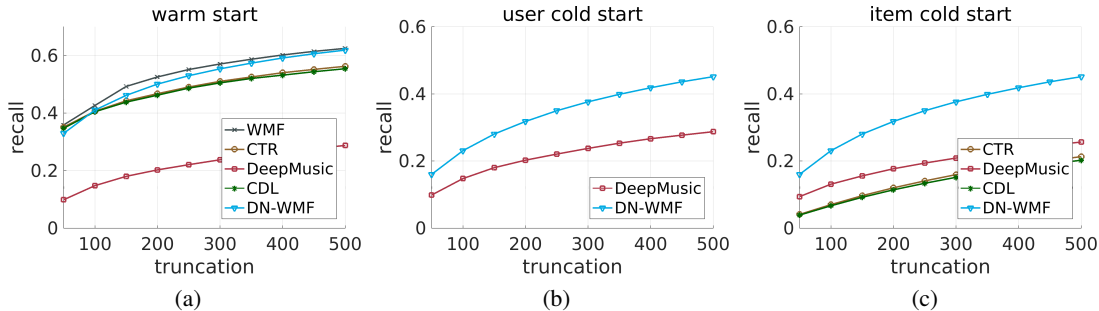

Figure 3: RecSys warm start (Figure 3(a)), user cold start (Figure 3(b)) and item cold start (Figure 3(c)) results. All figures show test set recall for truncations 50 to 500 in increments of 50. Code release by the authors of CTR and CDL is only applicable to item cold start so these baselines are excluded from user cold start evaluation.

location, title/tags, career level and other related information; see [2] for full description of the data. After cleaning and transforming all categorical inputs into 1-of-n representation we ended up with 831 user features and 2738 item features forming $\mathbf{\Phi}^{\mathcal{U}}$ and $\mathbf{\Phi}^{\mathcal{V}}$ respectively.

We process the interaction data by removing duplicate interactions (i.e. multiple clicks on the same item) and deletes, and collapse remaining interactions into a single binary matrix $\mathbf{R}$ where $\mathbf{R}(u, v) = 1$ if user $u$ interacted with job $v$ and $\mathbf{R}(u, v) = 0$ otherwise. We then split the data forward in time using interactions from the last two weeks as the test set. To evaluate both warm and cold start scenarios simultaneously, test set interactions are further split into three groups: warm start, user cold start and item cold start. The three groups contain approximately 426K, 159K and 184K interactions respectively with a total of $42,153$ cold start users and $49,975$ cold start items; training set contains 18.7M interactions. Cold start users and items are obtained by removing all training interactions for randomly selected subsets of users and items. The goal is to train a *single* model that is able to handle all three tasks. This simulates real-world scenarios for many online services like XING where new users and items are added daily and need to be recommended together with existing users and items. We set rank $D = 200$ for all models and in all experiments train our model (denoted DN-WMF) using latent representations from WMF. During training we alternate between applying dropout and inference approximation (see Section 4.3) for users and items in each mini-batch with a rate of 0.5. For CTR and CDL the code released by respective authors only supports item cold start so we evaluate these models on warm start and item cold start tasks only.

To find the appropriate DNN architecture we conduct extensive experiments using increasingly deeper DNNs. We follow the approach of [6] and use a pyramid structure where the network gradually compresses the input witch each successive layer. For all architecture we use fully connected layers with batch norm [14] and tanh activation functions; other activation functions such as ReLU and sigmoid produced significantly worse results. All models were trained using

| Network Architecture | Warm | User | Item |
|---|---|---|---|
| WMF | **0.426** | | |
| 400 | 0.421 | 0.211 | 0.234 |
| $800 \rightarrow 400$ | 0.420 | 0.229 | 0.255 |
| $800 \rightarrow 800 \rightarrow 400$ | 0.412 | **0.231** | **0.265** |

Table 2: Recsys recall@100 warm, user cold start and item cold start results for different DNN architectures. We use tanh activations and batch norm in each layer.

WMF as input latent model, however note that WMF cannot be applied to either user or item cold start. Table 2 shows warm start, user cold start, and item cold start recall at 100 results as the number of layers is increased from one to four. From the table we see that up to three layers, the accuracy on both cold start tasks steadily improves with each additional layer while the accuracy on warm start remains approximately the same. These results suggest that deeper architectures are highly useful for this task. We use the three layer model in all experiments.

RecSys results are shown in Figure 3. From warm start results in Figure 3(a) we see a similar pattern where all baselines perform comparably except DeepMusic, suggesting that content only models are unlikely to perform well on warm start. User and item cold start results are shown in Figures 3(b) and 3(c) respectively. From the figures we see that DeepMusic is the best performing baseline

significantly beating the next best baseline CTR on the item cold start. We also see that DN-WMF significantly outperforms DeepMusic with over 50% relative improvement for most truncations. This is despite the fact that DeepMusic was trained using the same 3-layer architecture and the same objective function as DN-WMF. These results further indicate that incorporating preference information as input into the model is highly important even when the end goal is cold start.

User inference results are shown in Figure 4. We randomly selected a subset of 10K cold start users that have at least 5 training interactions. Note that all training interactions were removed for these users during training to simulate cold start. For each of the selected users we then incorporate training interactions one at a time into the model in chronological order using the inference procedure outlined in Section 4.3. Resulting latent representations are tested on the test set. Figure 4 shows recall@100 results as number of interactions is increased from 0 (cold start) to 5. We compare with WMF by applying similar procedure from Equation 4 to WMF representations. From the figure it is seen that our model is able to seamlessly transition from cold start to preferences without retraining. Moreover, even though our model uses WMF as

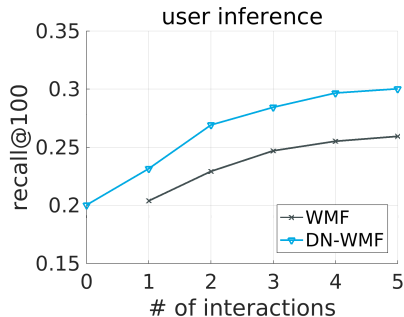

Figure 4: User inference results as number of interactions is increased from 0 (cold start) to 5.

input it is able to significantly outperform WMF at all interaction sizes. Item inference results are similar and are omitted. These results indicate that training with inference approximations achieves the desired effect allowing our model to transition from cold start to first few preferences without re-training and with excellent generalization.

## 6   Conclusion

We presented DropoutNet – a deep neural network model for cold start in recommender systems. DropoutNet applies input dropout during training to condition for missing preference information. Optimization with missing data forces the model to leverage preference and content information without explicitly relying on both being present. This leads to excellent generalization on both warm and cold start scenarios. Moreover, unlike existing approaches that typically have complex multi-term objective functions, our objective only has a single term and is easy to implement and optimize. DropoutNet can be applied on top of any existing latent model effectively, providing cold-start capabilities and leveraging full power of deep architectures for content modeling. Empirically, we demonstrate state-of-the-art results on two public benchmarks. Future work includes investigating objective functions that directly incorporate preference information with the aim of improving warm start accuracy beyond the input latent model. We also plan to explore different DNN architectures for both user and item models to better leverage diverse content types.

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
