[Reviews · NeurIPS 2017]

Reviewer 1



this paper proposed to use dropout for addressing cold-start problems in CF. The model proposed an architecture that uses user / item latent factors (can be learned from MF) and content features as input layer, and uses dropout on latent factors to make the model perform better on cold start scenarios. Overall the proposed method does not seem very novel: 1) dropout is a well known idea in neural networks, 2) take a step back, even before dropout / NN became popular, making sure training data / test data have similar distribution is common sense. Specifically, if we want to train a model to handle cold start scenarios, it is standard to not use user / item preferences. Also, from experiments it may not be necessary to use dropout to train a unified model at all. If we look at Figure 2, the performance on cold-start is highest with dropout = 1, then why not simply train a model without preferences (dropout = 1) to handle cold-start and train the full model to handle warm-start?

Reviewer 2



This paper presents a neural network based latent model that uses both the user / item responses and features for warm-start and cold-start scenarios. The main novelty claimed by the paper is to use drop-out to help better learn the feature-based latent model for cold-start. Below are some points: 1. Although it looks to me that leveraging drop-out in neural networks to better learn the feature-based model to help cold-start is a great idea, in this paper the methodology is not clearly written to be fully convincing. For example, in Algorithm 1, what is the probability of applying options 1-5 is not clearly stated. Is that a tuning parameter? If so, how does this relate to drop out rate? In Section 4.3, I am not sure if simply using mean(V_v) for user factors U_u is the best idea, some better justification is required there in my opinion. 2. Although the authors did a lot of experiments to demonstrate that this approach works better than the other baselines, there are a couple of points that I feel should be further improved. First, In Figure 3 (a) DNN-WMF was compared to a bunch of baselines, but in (b) it was only compared to DeepMusic, and in (c) WMF was not there. Was it intentional or there are some other reasons? Second, in Section 5.1 as the CiteULike is binary data, wouldn't it be better if we use a logistic Loss function rather than square d loss? 3. Some other minor points. (1) U_u and V_v were not properly defined in the paper. This caused a lot of confusion when I was reading it. (2) The paper needs a clear definition of cold-start vs warm-start. Does cold-start means no data at all for item / users? (3) Page 8, line 349, remove the duplicate "is able to".

Reviewer 3



Excellent paper, presenting an good idea with sufficient experiments demonstrating its efficiency. Well written and with a good survey of related work. The paper addresses the following problem: when mixing id level information (user id of item id) with coarser user and item features, any model has the tendency to explain most of the training data with the fine grained id features, using coarse features only to learn the rarest ids. As a result, the generated model is not good at inference when only coarser features are available (cold-start cases, of a new user or a new item). The paper proposes a dial to control how much the model balances out the fine grained and coarser information via drop-out of fine grained info while training. The loss function in Equation (2) has the merit of having a label value for all user-item pairs, by taking the output of the low rank model U_u V_v ^T as labels. My concern with such a loss is that the predictions from the low rank model have very different levels of accuracy: they are very accurate for the most popular users and the most popular items (users and items with most data), but can be quite noisy in the tail, and the loss function in Equation (2) does not account for the confidence in the low rank model prediction. Replacing the sum by a weighted sum using as weights some simple estimates of the confidence in each low rank model prediction could improve the overall model. But this can be done in future work. Question about experiments on CiteULike: Figure 2 and Table 1 well support the point the author is making about the effect of dropout on recall for both, warm start cases and cold start cases. However, it is surprising that the cold start recall numbers are better than the warm start recall numbers. Can the authors explain this phenomenon? Minor details: Line 138: f_\cal{U} -> f_\cal{V} Line 349: is able is able -> is able